# Phenotypic divergence between the cultivated apple (*Malus domestica*) and its primary wild progenitor (*Malus sieversii*)

**Thomas Davies, Sophie Watts, Kendra McClure, Zoë Migicovsky, Sean Myles** *

Department of Plant, Food, and Environmental Sciences, Faculty of Agriculture, Dalhousie University, Truro, NS, Canada

* sean.myles@dal.ca

**Data Availability Statement:** All data and code used for analysis are available from the GitHub repository: https://github.com/MylesLab/Wild_vs_cultivated.

## Abstract

An understanding of the relationship between the cultivated apple (*Malus domestica*) and its primary wild progenitor species (*M. sieversii*) not only provides an understanding of how apples have been improved in the past, but may be useful for apple improvement in the future. We measured 10 phenotypes in over 1000 unique apple accessions belonging to *M. domestica* and *M. sieversii* from Canada's Apple Biodiversity Collection. Using principal components analysis (PCA), we determined that *M. domestica* and *M. sieversii* differ significantly in phenotypic space and are nearly completely distinguishable as two separate groups. We found that *M. domestica* had a shorter juvenile phase than *M. sieversii* and that cultivated trees produced flowers and ripe fruit later than their wild progenitors. Cultivated apples were also 3.6 times heavier, 43% less acidic, and had 68% less phenolic content than wild apples. Using historical records, we found that apple breeding over the past 200 years has resulted in a trend towards apples that have higher soluble solids, are less bitter, and soften less during storage. Our results quantify the significant changes in phenotype that have taken place since apple domestication, and provide evidence that apple breeding has led to continued phenotypic divergence of the cultivated apple from its wild progenitor species.

## Introduction

The domesticated apple (*Malus domestica*) belongs to the genus *Malus*, which consists of 30–55 interfertile species that grow primarily in temperate climates. Archaeological evidence suggests that apples have been cultivated for at least 3,000 years [1] and that they have had immense cultural, religious, culinary and economic importance for centuries [2–4]. Genomic evidence suggests that as apples were transported west into Europe along the Silk Road from Central Asia, hybridization and introgression from multiple *Malus* species created the modern cultivated apple (*M. domestica*) [2, 5]. While there has been introgression from multiple species, including *Malus sylvestris* and *Malus baccata*, to the *M. domestica* genome, *Malus sieversii* of Kazakhstan is widely recognized as the primary ancestor of the cultivated apple [5–7].

**Funding:** This work was supported by the National Science and Engineering Research Council of Canada. ZM was supported by the National Science Foundation Plant Genome Research Program 1546869. The funders had no role in study design, data collection and analysis, decision to publish, or preparation of the manuscript.

**Competing interests:** The authors have declared that no competing interests exist.

Today, the cultivated apple is the 3rd most produced fruit crop in the world [8]. Accordingly, apple fruit quality and phenology traits have been a major focus for breeding programs around the world [9–11], and both wild and domesticated germplasm are routinely evaluated for their potential use by apple breeders [12, 13]. Traits such as precocity, harvest date and flowering date have practical implications for apple producers, as these traits influence investment timelines, crop quality and fruit damage risk. Weight, firmness, sugar content, acidity and phenolic content are important considerations for processors and consumers, who have specific preferences for these quality attributes when choosing to purchase apples [14]. Many of these fruit quality traits have been targets for improvement in breeding programs around the world, and current genetic mapping efforts remain focused on these phenotypes [15–17].

Cost-effective trait improvement in apples is critical since the investment costs of growing apple trees are high. Apple trees are large plants with a long juvenile phase: new trees often only start bearing fruit 5 years into the life cycle, requiring growers to invest heavily before generating revenue. Thus, producers typically grow only thoroughly evaluated and historically successful apple varieties. As a result, a small number of well-established varieties dominate the cultivated population. For example, in 2019 over 50% of all commercially produced apples in the US consisted of only 4 apple cultivars [18]. The global population of apples is dominated by a small number of elite varieties, despite an immense source of genetic and phenotypic diversity available for apple improvement [19]. Decreased diversity in apples, and agricultural crops more broadly, has resulted in an increased interest in the use of crop wild relatives (CWRs) for agricultural improvement. CWRs offer genetic and phenotypic diversity that can be leveraged in the breeding of novel cultivars with desirable traits such as disease resistance or flesh colour [20]. By 1997 the world economy had gained approximately $115 billion in benefits from the use of CWRs as sources of resistance to environmental change and disease [21]. An understanding of how fruit quality and phenology vary within the cultivated apple's wild relatives is essential to future apple improvement.

Phenotyping large and diverse populations of plants is labour intensive and frequently results in a "phenotyping bottleneck" [22], leaving crop researchers without powerful fruit quality data for analysis. Recently, comprehensive phenotyping of Canada's Apple Biodiversity Collection (ABC) generated measurements for fruit phenotypes in a collection of more than 1000 wild and cultivated apple accessions [23]. In the present work, we explored ten phenotypes from the ABC and determined the degree to which the cultivated apple differed from its primary wild progenitor, *M. sieversii*, and how cultivated apples have changed over the past 200 years of breeding and improvement.

## Materials and methods

### Phenotype data

The phenotype data analysed here were collected from Canada's Apple Biodiversity Collection (ABC) and were part of previously published work [23]. Briefly, the ABC is an apple germplasm collection located at the Agriculture and Agri-Food Canada (AAFC) Kentville Research Station in Nova Scotia, Canada (45.071767, -64.480466). The ABC contains 1119 unique accessions of apples planted in duplicate on M.9 rootstock in an incomplete randomized block design. The apple accessions in the ABC consist of accessions from the United States Department of Agriculture (USDA) Plant Genetic Resources Unit apple germplasm collection in Geneva, NY, USA; commercial cultivars from the Nova Scotia Fruit Growers' Association Cultivar Evaluation Trial; and diverse breeding material from AAFC Kentville. The orchard consists largely of *M. domestica* accessions, but also contains 78 *M. sieversii* accessions.

Phenotype data from the ABC were collected in 2016 and 2017 [23]. Here we focus on 10 phenotypes most relevant for assessing how apples have changed during domestication, breeding and improvement. Precocity was measured as a score of 1–4, indicating year of bloom; 1 (2014), 2 (2015), 3 (2016) and score 4 indicated that the tree had not yet bloomed as of 2016. Flowering date was measured in 2016 as the date in Julian days when the youngest wood displayed >80% of flowers at king bloom stage. Since it often took more than one day to harvest the entire orchard, harvest date was recorded in Julian days as the Monday of the week of harvest. Firmness was measured as the average firmness in kg/cm$^2$ at harvest of five apples measured using a penetrometer. Weight was measured as the average weight in grams of five apples at harvest. Acidity was measured as the malic acid content in mg/mL of combined juice from five apples measured using titration. Soluble solids were measured as˚Brix of the juice of five apples using a refractometer. Phenolic content was measured as μmol GAE/g of fresh weight. Percent acidity change was measured by subtracting the acidity at harvest from the acidity after 90 days storage and then dividing by the acidity at harvest. Percent firmness change was measured by subtracting the firmness at harvest from the firmness after 90 days storage and then dividing by the firmness at harvest. Sample sizes for each phenotype are listed in Table 1.

## Data analysis

Principal components analysis (PCA) was conducted using a scaled and centered matrix of the 10 phenotypes listed in Table 1 using the prcomp() function in R 4.0.2 [24]. A Wilcoxon signed-rank test was used to determine whether the phenotypes and PC values differed significantly between wild and cultivated apples.

A Pearson correlation was used to assess relationships between phenotypes and the release year of cultivated apples. Where appropriate, the significance threshold was Bonferroni-corrected to account for 10 comparisons. Data visualization was performed using the ggplot2 R package [25].

## Results

Sample sizes across the 10 phenotypes ranged from 9–76 and 399–797 for wild and cultivated apples, respectively, and are specified in Table 1. PCA of the 10 phenotypes revealed modest overlap between cultivated and wild apples in phenotypic space (Fig 1A and 1B). Wild and cultivated apples were significantly different along PC1 (W = 53893, p = 3.56 x 10$^{-26}$), PC2 (W = 13066, p = 2.07 x 10$^{-17}$) and PC3 (W = 39203, p = 0.0002; Fig 1C).

**Table 1. Sample sizes by phenotype.**

| Phenotype | M. domestica | M. sieversii |
|---|---|---|
| Precocity | 797 | 76 |
| Flowering Date | 768 | 74 |
| Harvest Date | 647 | 59 |
| Firmness | 644 | 59 |
| Weight | 644 | 58 |
| Acidity | 626 | 56 |
| Soluble Solids | 644 | 56 |
| Phenolic Content | 399 | 9 |
| % Change in acidity during storage | 449 | 19 |
| % Change in firmness during storage | 409 | 27 |

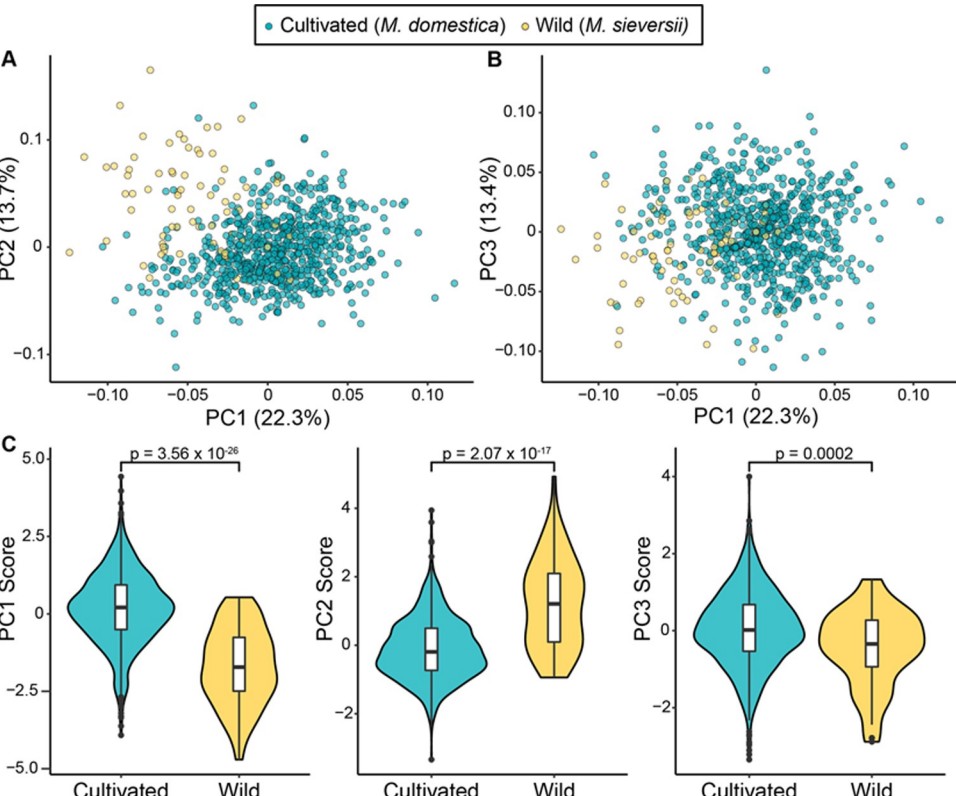

**Fig 1. PCA of ten phenotypes in wild (N = 79) and cultivated apples (N = 801).** A) PC1 vs PC2. B) PC1 vs PC3. The proportion of the variance explained by each PC is shown in parentheses on each axis. C) The difference between wild and cultivated apples for PCs 1, 2 and 3 are shown as violin plots. P values from a Wilcoxon test comparing PC values between cultivated and wild apples are shown for each of the first three PCs.

To visualize and assess the difference between cultivated and wild apples for each individual phenotype, we produced density plots to visualize each species' distribution for each phenotype and tested whether phenotypes differed between the two species (Fig 2).

Wild and cultivated apples differed significantly for 6 of the 10 phenotypes tested, including precocity (W = 23838, p = 0.021), flowering date (W = 48984, p = $7.52 \times 10^{-24}$), harvest date (W = 30482, p = $2.99 \times 10^{-13}$), weight (W = 36255, p = $1.44 \times 10^{-31}$), acidity (W = 8480, p = $5.1 \times 10^{-9}$), and phenolic content (W = 352, p = $5.59 \times 10^{-5}$). We found that, on average, cultivated apples produce flowers for the first time 21% (0.38 years) earlier than wild apples. Within a growing season, cultivated apples flower 3 days later, and are harvested 15 days later than wild apples. Cultivated apples are also 3.6 times heavier, 43% less acidic, and 68% lower in phenolic content than their wild progenitors. In comparison, wild and cultivated apples did not differ significantly for firmness, soluble solids, or changes in acidity or firmness during storage.

To visualize phenotypic change within cultivated apples over time, apples' phenotypes are displayed as a function of their release year (Fig 3 & S1 Fig). We found significant correlations with release year for phenolic content (R = -0.364, p = $2.34 \times 10^{-6}$), change in firmness during storage (R = 0.222, p = 0.00265), flowering date (R = -0.172, p = 0.00247), and soluble solids (R = 0.123, p = 0.0469) and determined that cultivated apples have shifted closer to the mean of wild apples for flowering date and firmness change, but further from the mean of wild apples for phenolic content and soluble solids.

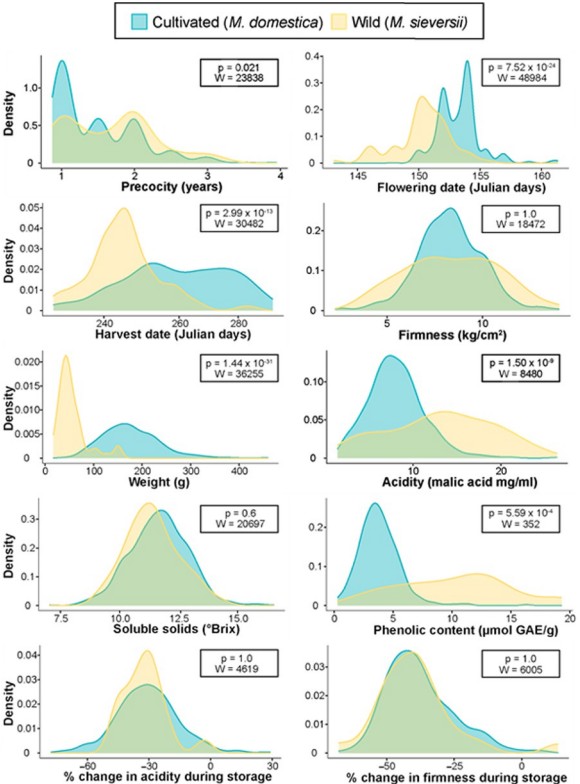

**Fig 2. Overlapping density plots of 10 phenotypes comparing values from wild and cultivated apples.** The phenotype associated with each plot is shown along the X axis. The W and Bonferroni-corrected p values report the results of performing a Wilcoxon rank sum test of the difference between the phenotypic distributions of wild and cultivated apples.

## Discussion

Apples have been cultivated for over 3000 years, but because vegetative propagation has been practiced for 2000 years, it has been suggested that only about 100 generations have elapsed since apple domestication [26]. Despite this relatively short window for apple improvement, we found that cultivated apples are nearly entirely phenotypically distinct from their primary wild progenitor, *M. sieversii* (Fig 1). Phenotypic differences are frequently used as an approximate measure of relatedness, and the separation in principal component space observed here is in agreement with genomic studies that have shown significant differentiation between the genomes of *M. domestica* and *M. sieversii* [5, 19]. It is worth acknowledging that we observed some overlap between wild and cultivated apples in phenotypic space. The PCA performed here made use of only 10 phenotypes, and it is possible that more differentiation would be observed with more measures of the apple phenome. Further, each variable in PCA should ideally capture an independent biological feature of apples. However, some phenotypes analysed here are correlated, such as harvest date and firmness [23], and their variation may be driven by the same biological feature [27]. Therefore, interpreting our PCA as a quantification of the degree of phenotypic differentiation between cultivated and wild apples should take these caveats into consideration.

We found significant differences between wild and cultivated apples for several phenology traits including precocity, flowering date, and harvest date (Fig 2). Cultivated apple trees flower and bear fruit at a younger age. Due to the long juvenile phase of apple trees, plants with the

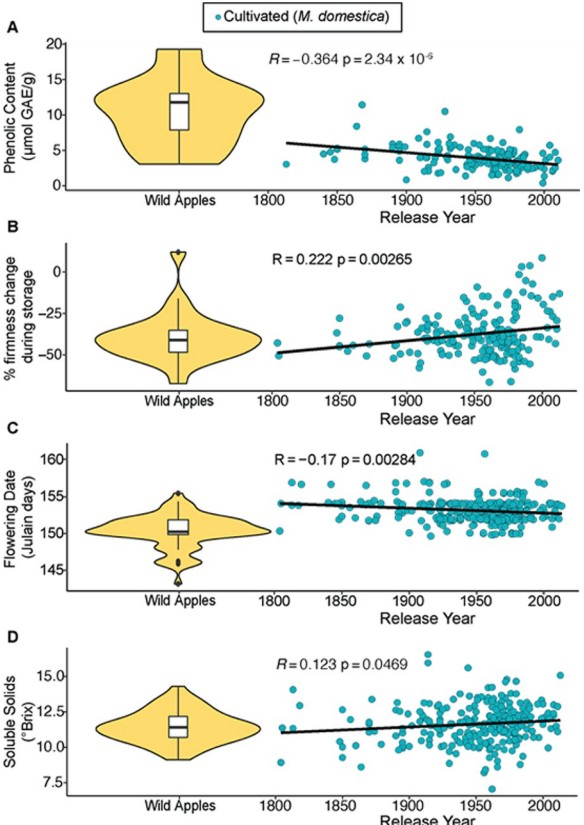

**Fig 3. Phenotype values of cultivated apples as a function of their release year with a comparison to values in their wild ancestor, *M. sieversii*.** Phenotypes include phenolic content (A), firmness change during storage (B), flowering date (C), and soluble solids (D). Values for cultivated apples are blue, and the values observed for *M. sieversii* are represented in yellow as a violin plot on the left side of each plot. The R and p values from a Pearson correlation between phenotypic values and release year are shown within each scatter plot.

ability to bear fruit earlier in their life cycle are desirable for growers because revenue is generated earlier. It is therefore possible that precocity has been selected for during apple improvement.

Flowering date was 17% (3 days) later in cultivated apples than wild apples. Frost during blossoming can cause loss, damage or reduced marketability of fruits [28], making flowering time an important consideration for growers when planting orchards. Additionally, apples with later flowering dates tend to be firmer [23, 29], and firmer apples are preferred by consumers [30]. The later flowering date in cultivated apples could therefore be a by-product of selection for firm apples. Similarly, selection for firm apples may explain why cultivated apples were harvested 15 days later than wild apples, since harvest date and firmness are strongly correlated [23, 29]. It is well established that harvest date is a reliable predictor of fruit firmness, and these two phenotypes may be regulated by a common molecular pathway [27]. Thus, preference for firm fruit could be directly impacting the selection for apples with later harvest dates.

We found significant differences between cultivated and wild apples across multiple fruit traits including weight, acidity, and phenolic content (Fig 2). Cultivated apples are 3.6x heavier than wild apples, in agreement with previous comparisons between these two species [31]. Consumers prefer large, visually appealing fruit [32, 33], so selection for large fruit size may

explain our observation. We also found that cultivated apples are 43% less acidic than wild counterparts. Acidity contributes to the sour taste of apples, and apple preference is heavily influenced by acid/sugar ratios [34]. Given this relationship, it is not surprising that cultivated apples, which are primarily consumed as fresh fruit [35], have lower acid than wild apples but do not differ in soluble solid content. Finally, cultivated apples have, on average, 68% less phenolic content than wild apples. Phenolic compounds, which offer nutritional benefits [36], are partially responsible for the enzymatic browning that occurs when apple flesh is exposed to oxygen [37]. Browned flesh is visually unappealing and typically results in negative effects on flavour, making apples that resist browning more appealing to producers and consumers [37]. In fact, the only genetically modified apple variety on the market today, Arctic™ Apples, was designed to silence genes related to enzymatic browning and was advertised as "the original nonbrowning apple" [38]. The human aversion to apple browning has likely contributed to the decline in phenolic content in cultivated apples, despite the nutritional benefits of such compounds. In addition, some evidence suggests that fruit size impacts polyphenol accumulation in apples [39], which could help explain why we observe lower phenolic content in cultivated apples.

According to the present analysis, many phenotypes of cultivated apples have dramatically changed since divergence from the primary progenitor species, *M. sieversii*. These differences represent phenotypic separation that could be leveraged in the improvement of cultivated apples, and emphasizes the potentially functional diversity provided by CWRs. While wild apples from this investigation may not offer improved fruit quality phenotypes that are currently attractive to consumers, they hold phenotypic variation that could be important for apple improvement in the future. For example, breeders could exploit the high phenolic content of wild apples to improve the nutritional quality of cultivated apples. Further, traits from wild apple varieties could potentially benefit the cider industry, which values high acidity and phenolic content [40].

Analysis of cultivated apple phenotypes as a function of release year revealed changes over the past 200 years in phenolic content, change in firmness during storage, flowering date, and soluble solids (Fig 3). In particular, as shown previously [23], phenolic content has decreased over time. Phenolic content is associated with bitter taste [41], and modern varieties therefore likely taste less bitter on average than older varieties. Although selection for decreased bitterness could explain our observation, the relationship between low phenolic content and decreased flesh browning could also explain why modern cultivated apples tend to have less phenolics [42]. In comparison, wild apples tend to have higher phenolic content, indicating that cultivated varieties are diverging from the ancestral state. Similarly, more recently released apple cultivars soften less during storage than older cultivars, diverging from the ancestral state. The extended storage and long-distance shipment of apples has become increasingly routine over the past several decades, and selection for reduced softening during storage may explain why firmness retention has improved over time. Storage and transport have also been key targets in tomato breeding [43], and the demand for fruit that performs well during extended storage and transport is unlikely to subside.

Flowering date is an important trait for apple production, and varies widely across the genus *Malus* [13]. Later flowering apple trees are less likely to be impacted by frost damage [28] and more likely to be firm [23], which is preferred by consumers. Despite the understood benefits of growing apples with later flowering dates, we found that more recently released varieties had earlier flowering dates. The trend towards earlier flowering varieties could indicate that selection for other traits has indirectly impacted flowering date. Alternatively, growers could be preferring earlier flowering varieties in an attempt to manage fruit ripening times during the harvest season. Cultivated varieties are trending towards the ancestral state of

earlier flowering dates, which suggests that wild apples could offer valuable genetic material for breeding earlier harvested varieties.

Finally, we found that more modern cultivated apples are only slightly higher in soluble solid content. Previous investigations have reported that firm apples tend to have higher sugar content [10, 29, 44], so our observation that modern apple varieties tend to have higher SSC may be at least partially be driven by recent selection for increased firmness. Further, a number of studies have suggested that the sugar content of apples is a key factor affecting consumer preference [14, 30]. Although SSC is only a modest predictor of perceived sweetness [45], consumer's preference for sweet apples could underlie the upward trend in soluble solid content seen in modern cultivated apples.

Several caveats of the present analysis are worth noting. First, we only considered one of the multiple progenitor species of *M. domestica* here [6]. Therefore, only a fraction of the ancestry of the cultivated apple is captured by *M. sieversii*, and a more inclusive pool of ancestral species would yield a more comprehensive comparison of wild and cultivated apples. Second, it is unknown how representative the current sample of wild apples is of the broader *M. sieversii* population. It is possible that the wild apple varieties within the ABC represent only an unrepresentative subset of *M. sieversii*, and thus do not accurately capture the diversity of the species. Further, there has been evidence of gene flow between cultivated and wild apples [46], which could mean that the wild species from the current investigation have experienced gene introgression from cultivated trees, and thus do not accurately represent the wild progenitor. Finally, the relatively small sample size in several comparisons limited the power of some of our analyses (Table 1).

Our work demonstrates that cultivated and wild apples have diverged phenotypically, and that hundreds of years of apple improvement have shaped the variation in fruit and phenology we observe among cultivated apples today. Wild apples offer potentially valuable pools of genetic material that may be helpful for apple improvement. Future holistic evaluations including a combination of genomic, metabolomic and transcriptomic analyses, will help further assess the degree to which the apple's wild relatives may contribute to improving apple cultivar development.

## Supporting information

**S1 Fig. Phenotypes of cultivated apples as a function of their release year with a comparison to the ancestral state.** Phenotypes include acidity change during storage, acidity, precocity, harvest date, firmness, and weight. Cultivated apple scores for each phenotype are shown in blue, and the ancestral state of each phenotype is represented in yellow as a density distribution of values from *M. sieversii*. The R and p values from a Pearson correlation between phenotypic values and release year are shown within each scatter plot.
(DOCX)

## Acknowledgments

**General:** The authors thank the Nova Scotia Fruit Growers' Association and the Farm Services team at AAFC-Kentville for their work in establishing and maintaining the trees studied here. We thank Tayab Soomro for useful discussion.

## Author Contributions

**Conceptualization:** Thomas Davies, Zoë Migicovsky, Sean Myles.

**Data curation:** Thomas Davies, Sophie Watts, Kendra McClure, Zoë Migicovsky.

**Formal analysis:** Thomas Davies.

**Funding acquisition:** Sean Myles.

**Investigation:** Sean Myles.

**Methodology:** Sophie Watts, Kendra McClure, Sean Myles.

**Project administration:** Sean Myles.

**Supervision:** Zoë Migicovsky, Sean Myles.

**Visualization:** Thomas Davies.

**Writing – original draft:** Thomas Davies.

**Writing – review & editing:** Thomas Davies, Sophie Watts, Kendra McClure, Zoë Migicovsky, Sean Myles.

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
