## [Decision Letter · Decision Letter 0]

31 May 2021

PONE-D-21-11607

Phenotypic divergence between the cultivated apple (Malus domestica) and its primary wild progenitor (Malus sieversii)

PLOS ONE

Dear Dr. Myles,

Thank you for submitting your manuscript to PLOS ONE. After careful consideration, we feel that it has merit but does not fully meet PLOS ONE’s publication criteria as it currently stands. Therefore, we invite you to submit a revised version of the manuscript that addresses the points raised during the review process.

We look forward to receiving your revised manuscript.

Kind regards,

Mohar Singh, Ph.D. Plant Breeding

Academic Editor

PLOS ONE

Additional Editor Comments:

All suggested changes need to be incorporate

Journal Requirements:

4. Thank you for stating the following in the Funding Section of your manuscript:

[This work was supported by the National Science and Engineering Research Council of Canada. ZM was supported by the National Science Foundation Plant Genome Research Program 1546869. ]

 [The funders had no role in study design, data collection and analysis, decision to publish, or preparation of the manuscript.]

Reviewers' comments:

Reviewer's Responses to Questions

**Comments to the Author**

1. Is the manuscript technically sound, and do the data support the conclusions?

Reviewer #1: Yes

Reviewer #2: Yes

2. Has the statistical analysis been performed appropriately and rigorously? 

Reviewer #1: Yes

Reviewer #2: Yes

3. Have the authors made all data underlying the findings in their manuscript fully available?

Reviewer #1: Yes

Reviewer #2: Yes

4. Is the manuscript presented in an intelligible fashion and written in standard English?

Reviewer #1: Yes

Reviewer #2: Yes

5. Review Comments to the Author

Reviewer #1: Comments:

The article is well structured and nicely organized. It explores the phenotypic divergence between the cultivated apple and its primary wild progenitor. Authors have shown that the cultivated apple has lower acidity and less phenolic content than the wild one. The study has provided an in-depth coverage for variation in fruit and phenology across both species which may help in improving future apple cultivars.

The article could be accepted for publication if following minor revisions are incorporated:

In discussion part, add a holistic approach including phenomics, genomics and metabolomics, which could be utilized for apple improvement programs.

Correct the grammatical mistakes and sentence making. It will be better if help of some native English speaker could be taken.

References need thorough cross-checking. Please check for journal abbreviations. Provide wherever applicable and avoid unnecessary ones, if not available.

Regards

Reviewer #2: Observations:

Introduction –

The authors have written introduction very nicely with covering the importance of genetic variation with respect to cultivated and wild apple progenitor.

Materials and Methods –

1.This section has been written systematically.

2.Please specify the number of apple accessions evaluated for ten phenotypes. Is the present study involving data from 1119 ABC.

Results

1.The results obtained by the authors are very interesting.

2.Results indicated the correlation of selected trait in different accessions with respect to year to varietal release and phenological stages.

Discussion and Conclusion

The present research is justified with good sources and presented in a well form.

Overall Recommendation – Recommended for publication.

6. PLOS authors have the option to publish the peer review history of their article (what does this mean?). If published, this will include your full peer review and any attached files.

Reviewer #1: No

Reviewer #2: **Yes: **Dr Narender Negi

Scientist, Fruit Science

ICAR-National Bureau of Plant Genetic Resources,

Regional Station, Phagli-Shimla,

Himachal Pradesh, 171 004.

Cell No: +91 9418317335

---

## [Editor Report · Decision Letter 1]

22 Feb 2022

Phenotypic divergence between the cultivated apple (Malus domestica) and its primary wild progenitor (Malus sieversii)

PONE-D-21-11607R1

Dear Sir,

We’re pleased to inform you that your manuscript has been judged scientifically suitable for publication and will be formally accepted for publication once it meets all outstanding technical requirements.

Kind regards,

Mohar Singh, Ph.D. Plant Breeding

Academic Editor

PLOS ONE
---

## [Editor Report · Acceptance letter]

28 Feb 2022

PONE-D-21-11607R1 

Phenotypic divergence between the cultivated apple (*Malus domestica*) and its primary wild progenitor (*Malus sieversii*) 

Dear Dr. Myles:

I'm pleased to inform you that your manuscript has been deemed suitable for publication in PLOS ONE. Congratulations! Your manuscript is now with our production department. 

Kind regards, 

on behalf of

Dr. Mohar Singh 

Academic Editor

PLOS ONE